# Multiple independent losses of crossover interference during yeast evolutionary history

**Abhishek Dutta**[1], **Fabien Dutreux**[1], **Marion Garin**[1], **Claudia Caradec**[1], **Anne Friedrich**[1], **Gauthier Brach**[1], **Pia Thiele**[1], **Maxime Gaudin**[2], **Bertrand Llorente**[2]*, **Joseph Schacherer**[1,3]*

**1** Université de Strasbourg, CNRS, GMGM UMR7156, Strasbourg, France, **2** CNRS UMR7258, INSERM U1068, Aix Marseille Université UM105, Institut Paoli-Calmettes, CRCM, Marseille, France, **3** Institut Universitaire de France (IUF), Paris, France

* bertrand.llorente@inserm.fr (BL); schacherer@unistra.fr (JS)

**Data Availability Statement:** Sequence data are available from NCBI sequence read archive under accession number SRP PRJEB71080 and PRJEB77073.

## Abstract

Meiotic recombination is essential for the accurate chromosome segregation and the generation of genetic diversity through crossover and gene conversion events. Although this process has been studied extensively in a few selected model species, understanding how its properties vary across species remains limited. For instance, the ancestral ZMM pathway that generates interference-dependent crossovers has undergone multiple losses throughout evolution, suggesting variations in the regulation of crossover formation. In this context, we first characterized the meiotic recombination landscape and properties of the *Kluyveromyces lactis* budding yeast. We then conducted a comprehensive analysis of 29,151 recombination events (19, 212 COs and 9, 939 NCOs) spanning 577 meioses in the five budding yeast species *Saccharomyces cerevisiae*, *Saccharomyces paradoxus*, *Lachancea kluyveri*, *Lachancea waltii* and *K. lactis*. Eventually, we found that the *Saccharomyces* yeasts displayed higher recombination rates compared to the non-*Saccharomyces* yeasts. In addition, bona fide crossover interference and associated crossover homeostasis were detected in the *Saccharomyces* species only, adding *L. kluyveri* and *K. lactis* to the list of budding yeast species that lost crossover interference. Finally, recombination hotspots, although highly conserved within the *Saccharomyces* yeasts are not conserved beyond the *Saccharomyces* genus. Overall, these results highlight great variability in the recombination landscape and properties through budding yeasts evolution.

## Author summary

Meiotic recombination ensures proper chromosome segregation and promoting genetic diversity. Studies in *Saccharomyces cerevisiae* have provided mechanistic insights into the process of meiotic recombination, but the evolutionary diversity of the meiotic recombination pathway itself is not well understood. Here, we examined recombination in the non-model yeast *Kluyveromyces lactis* and compared it with four closely related budding yeast species—*S. paradoxus*, *L. waltii*, *L. kluyveri*, *S. cerevisiae*—to explore the evolution of the crossover pathway. We observed significant variation in recombination rates among

**Funding:** Agence Nationale de la Recherche (ANR): JS ANR-18-CE12-0013; Agence Nationale de la Recherche (ANR):BL ANR-18-CE12-0013; EC | European Research Council (ERC):JS 772505 The funders had no role in study design, data collection and analysis, decision to publish, or preparation of the manuscript.

**Competing interests:** The authors have declared that no competing interests exist.

these species alongside the loss of the crossover interference pathway. While crossover assurance is essential in *S. cerevisiae* and *S. paradoxus*, it seems defective in non-*Saccharomyces* yeasts, suggesting alternative mechanisms ensuring faithful meiosis. Additionally, recombination hotspots are conserved only within *Saccharomyces* yeasts but not beyond, highlighting considerable variability in the recombination landscape across budding yeast evolution.

## Introduction

In conjunction with sister chromatid cohesion, crossovers (CO) between homologous chromosomes during meiotic prophase ensure physical connections facilitating accurate chromosome segregation [1]. Failure to establish at least one CO per homolog pair increases the probability of homolog nondisjunction at the first meiotic division, resulting in aneuploid gametes [2].

Meiotic CO result from the repair by homologous recombination of programmed DNA double strand breaks (DSB) catalyzed by the evolutionarily conserved topoisomerase-like protein Spo11 [3–5]. Spo11-DSB-induced meiotic recombination also yields non-crossovers (NCO), mainly through a distinct pathway than the one yielding CO [6–8]. Eventually, meiotic recombination also yields genetically silent products when the repair occurs between sister chromatids, although the repair is strongly biased toward the use of the homolog [9]. In many organisms including the well-studied budding yeast *Saccharomyces cerevisiae*, Spo11-DSBs and subsequent recombination events are enriched in hotspots, and crossover hotspots are well conserved in the *Saccharomyces* genus [10].

CO formation is tightly regulated by various mechanisms collectively referred to as CO control, that involves CO interference, CO homeostasis and CO assurance [11,12]. Interference regulates CO patterning along chromosomes, where the presence of a CO prevents the formation of another CO in its vicinity. This ensures a rather even distribution of COs along chromosomes [13–17]. CO homeostasis refers to the phenomenon maintaining COs at the expense on NCOs when Spo11-DSBs become limiting [11,18]. CO assurance refers to the process that facilitates the formation of at least one CO per homolog pair [19]. Organisms exhibiting strong interference, such as *Caenorhabditis elegans*, *Drosophila*, and humans, ensure a single CO per homolog pair, even when the total number of COs per meiosis is limited. In contrast, organisms with weaker interference, like *S. cerevisiae*, necessitate a considerably higher number of CO per meiosis to ensure at least one CO per homolog pair [20,21]. Although the molecular bases of these mechanisms are still poorly understood, CO homeostasis seems to result from CO interference, but CO assurance seems independent of them and established upstream [22].

Interfering CO, also known as class I CO, are produced via the evolutionarily conserved ZMM pathway, historically composed of Zip1-4, Spo16, Msh4/5, and Mer3 in *S. cerevisiae* [23–27]. The ZMM pathway relies on the nuclease activity of the Mlh1-Mlh3 heterodimer to exclusively resolve recombination intermediates into CO [28–30]. Except organisms like *C. elegans* where only class I CO exist; non-interfering class II CO also co-exist with class I CO but are generally less abundant. They form independently of the ZMM proteins through the resolution of recombination intermediates by the structure-specific nucleases Mus81, Yen1, and Slx1 [31–35]. Interestingly, in organisms lacking the ZMM pathway such as the fission yeast *Schizosaccharomyces pombe* and the budding yeast *Lachancea waltii*, CO also show

interference, but its extent is rather limited. The most likely explanation is that such CO interference signal results from the non-random patterning of CO precursors [17,36–40].

Recombination and shuffling of homologous chromosomes during meiosis serve as significant contributors to genetic diversity, underscoring the crucial role of meiosis in species evolution [41]. However, the evolutionary dynamics of recombination rates is poorly understood. Despite the tight regulation, variability in recombination rates has been observed at multiple levels, including within and between chromosomes, individuals, sexes, populations, and species [42]. Analyzing recombination landscapes among closely related species can offer valuable insights into the evolutionary forces influencing recombination rate variation. The comparisons of fine-scale recombination maps can make it possible to identify features influencing CO formation, patterning, and control as well as facets impacting shifts in recombination landscapes between species. In this context, we previously characterized the recombination landscapes of two species of the *Lachancea* clade, *Lachancea kluyveri* and *Lachancea waltii*, and highlighted several differences with *S. cerevisiae* [39,43]. *Lachancea* recombination rates are overall lower than that of *S. cerevisiae*, and the incidence of non-exchange chromosomes (E0 chromosomes) is much higher in *Lachancea* species. We also revealed a complete absence of recombination on the entire chromosome arm containing the sex locus in *L. kluyveri* due to the lack of chromosome axis protein recruitment [43,44]. Finally, in agreement with the loss of Zip2, 3 and 4, Spo16, Msh4 and 5 right after the divergence from *L. kluyveri* [45], we showed that *L. waltii* exhibits only minimal CO interference likely resulting from CO precursors patterning [39].

Here, we characterized the meiotic recombination landscape of the budding yeast species *Kluyveromyces lactis*. As the *Lachancea* species, *K. lactis* diverged from *S. cerevisiae* prior to the whole genome duplication event that characterizes this latter lineage [46]. Integration of *K. lactis* meiotic recombination properties in a broad picture composed of five budding yeast species of similar genome sizes—*S. cerevisiae*, *Saccharomyces paradoxus*, *L. kluyveri*, *L. waltii*, and *K. lactis*—revealed substantial variability between *Saccharomyces* and non-*Saccharomyces* species. Non-*Saccharomyces* species show reduced rates of both CO and NCO, high incidence of non-exchange chromosomes and only residual crossover interference. Furthermore, while recombination hotspots are highly conserved within the *Saccharomyces* genus, they do not display conservation across the broad evolutionary distances explored in our study.

## Results

### Generation of a population of 820 *Kluyveromyces lactis* segregants

A way to explore the recombination landscape is to genotype the meiotic progeny of an intraspecies hybrid. The *K. lactis* species is characterized by a high level of genetic diversity (S1 Fig) and post-zygotic reproductive isolation [47,48]. The overall genetic diversity of this species is high ($\theta_w = 3.3 \times 10^{-2}$) compared with other species such as *S. cerevisiae* ($\theta_w = 1.6 \times 10^{-2}$). By testing different pairs of isolates, we found that the cross between UCD 67–376 (C6) and UCD 72–212 (H7), two isolates from the Asian clade, results in a viable diploid hybrid with good enough spore viability (96% viable spores), suitable for generating a segregating population.

The genomes of *K. lactis* natural isolates present numerous structural variants compared to the CBS2359 reference genome [49]. To generate high quality genome assemblies and detect the structural variants (SVs) of the C6 and H7 strains, we sequenced their genomes using both long-read and short-read sequencing strategies. We successfully assembled the two *K. lactis* genomes (10.9 Mb) into seven scaffolds representing six chromosomes (scaffolds S1–6, Fig 1; S1A Table) and the mitochondrial genome (S1A Table, S2A–S2B Fig). We detected a total of 133 and 141 SVs in the genomes of C6 and H7, respectively, compared to the reference isolate.

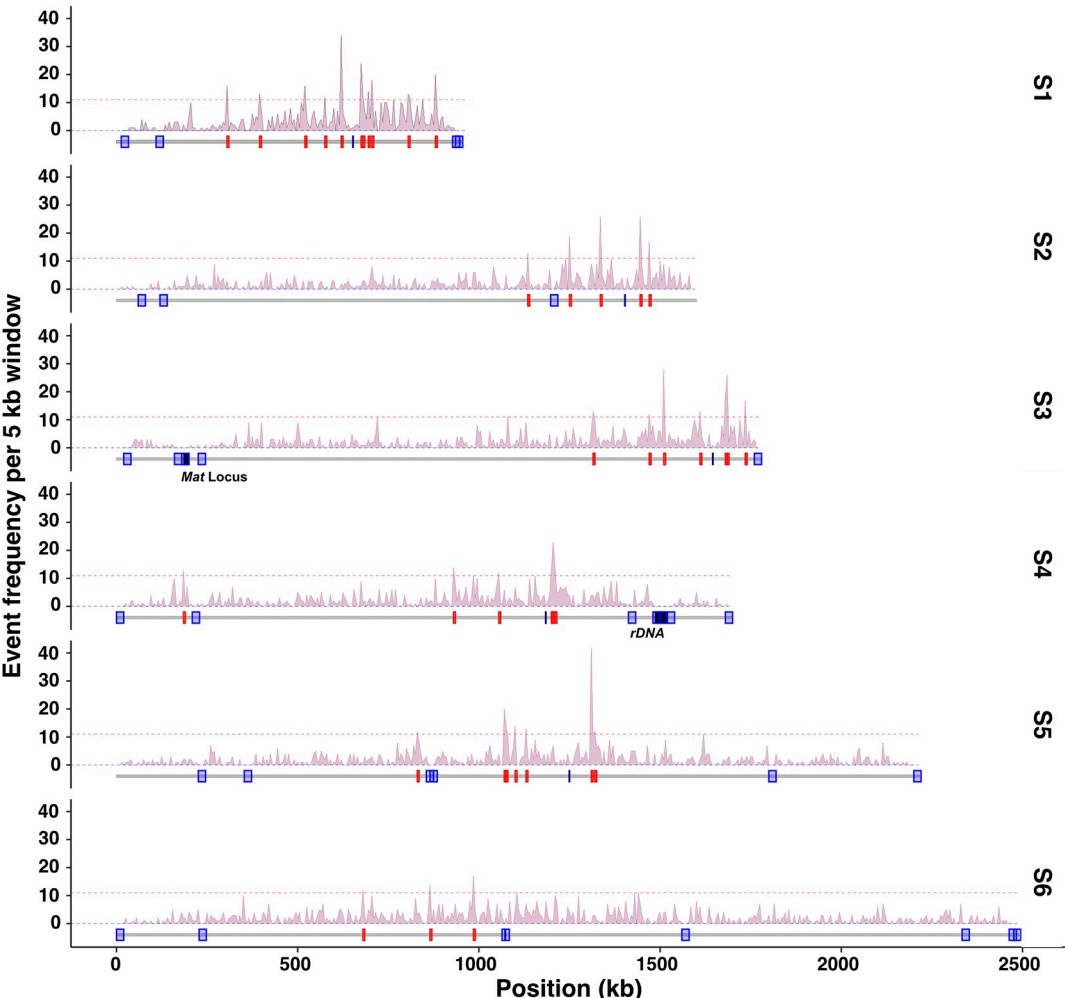

**Fig 1.** *Kluyveromyces lactis* **recombination map.** Density of CO along the six chromosomes represented by scaffolds S1-6 using a 5 kb window. Horizontal dashed red lines represent the CO hotspot significance threshold (see Materials and methods). Under the density plot, CO coldspots and hotspots are shown in blue and red, respectively. For each chromosome the black dash represents the centromere position. The *MAT* locus on chromosome 3 and the rDNA locus on chromosome 4 are highlighted in black.

These variants included deletions, insertions, duplications, and contractions of repetitive sequences as well as rearrangements such as inversions and translocations (S3A–S3B Fig, S1B Table). However, the genomes of the C6 and H7 isolates were found to be globally collinear as expected from the high spore viability of the corresponding hybrid (S3C Fig). Eventually, we used these new genome assemblies in all subsequent analyzes.

We selected 29,817 reliable Single Nucleotide Polymorphisms (SNPs) as genetic markers along the chromosomes of the C6 and H7 isolates. This corresponds to a genetic divergence of ~0.3 SNP per 100 nucleotides (0.3%) and a median inter-marker interval size of 187 bp (S4 Fig). This genetic divergence is between two to four fold lower than that of other yeast hybrids previously used to dissect recombination landscapes, but it is suitable to generate a comprehensive recombination map (Table 1) [39,43,50,51]. Finally, we sequenced a population of 820 segregants (*i.e.*, 205 four-spore viable tetrads) from the C6/H7 hybrid (Figs 1 and S1B). Compared to what was previously observed in *Lachancea* species, the *K. lactis* segregating

**Table 1. Comparison of hybrid DNA sequence divergence (#SNP/100bp), meiotic spore viability, number of tetrads analyzed, total and median number of CO and NCO events across the species.** Note that because of LOH blocks present in the hybrids of *L. waltii* and *L. kluyveri*, the total CO and NCO counts are underestimates. The median values have been corrected for the non-informative LOH fraction of the genome.

| Species | Hybrid divergence | Spore viability | Tetrads | Total CO (median/tetrad) | Total NCO (median/tetrad) |
|---|---|---|---|---|---|
| *S. cerevisiae* | 0.5% | 85% | 72 | 6659 (93) | 3648 (45) |
| *S. paradoxus* | 1.1% | 64% | 49 | 2766 (56) | 1540 (30) |
| *K. lactis* | 0.3% | 96% | 205 | 4702 (23) | 2947 (14) |
| *L. waltii* | 0.6% | 71% | 192 | 4049 (34) | 1459 (12) |
| *L. kluyveri* | 0.5% | 70% | 59 | 1110 (19) | 345 (6) |

population was devoid of scars indicating aneuploidies or loss of heterozygosity (LOH) events in the diploid progenitor [39,43].

## *Kluyveromyces lactis* recombination landscape and CO interference

Across the 205 meioses, we identified 4,702 COs and 2,947 NCOs. This corresponds to a median of 23 COs and 14 NCOs per meiosis (Fig 1, Table 1), which is equivalent to 3.8 COs and 2.3 NCO per chromosome and a rate of 2.3 COs and 1.6 NCOs per Mb (Fig 2A). We observed that 65% of the COs were accompanied by detectable gene conversion tracts. The median conversion tract size associated with COs and NCOs is 1.8 kb and 1.2 kb, respectively, comparable to previous estimates in other budding yeast species (S5 Fig). The average frequency of COs and NCOs showed a linear relationship with chromosome size, but unlike in *S. cerevisiae*, the CO density was not inversely correlated with chromosome size (Figs 2B and S6). Finally, 11.2% of the tetrads have one non-exchange or $E_0$ (0 for no exchange) chromosome, and one tetrad has more than one non-exchange chromosome (S2 Table). CO homeostasis manifests itself by a negative correlation between the CO:NCO ratio and the total number of recombination events (CO+NCO) per tetrad. We observed no correlation between the CO:NCO ratio and CO+NCO in *K. lactis*, hence no CO homeostasis (Fig 3).

The ZMM pathway is present in most eukaryotes, including humans, plants, and *S. cerevisiae*, although it has been independently lost several times during evolution, particularly in yeasts [39,45,52]. Interestingly, all the genes of the ZMM pathway were detected *in K. lactis*, except *SPO16*. In addition, a premature stop codon resulting from a frameshift was detected in *MSH5*, most likely leading to a loss of function. These properties question the functionality of the ZMM pathway, hence the presence of CO interference in *K. lactis*.

We analyzed CO interference first using the coefficient of coincidence (CoC) method, and second, using a gamma distribution to model inter-CO distances [39,50,53]. The coefficient of coincidence measures the actual frequency of double CO occurring in any pair of chromosomal intervals to the expected frequency of such double CO if they were generated independently in each interval. It quantifies the deviation from independent CO events between the intervals. This analysis shows that in *K. lactis* the CoC is positive over very short inter-interval distances, similar to *L. waltii* (Fig 4). By using a gamma distribution to model inter-CO distances, it is possible to get a quantitative measure of CO interference. A value of $\gamma = 1$ indicates the absence of interference, while $\gamma > 1$ means the presence of positive interference. For *K. lactis*, the $\gamma$-value was 1.27, which is significantly different from a distribution with $\gamma = 1$ (S7 Fig)

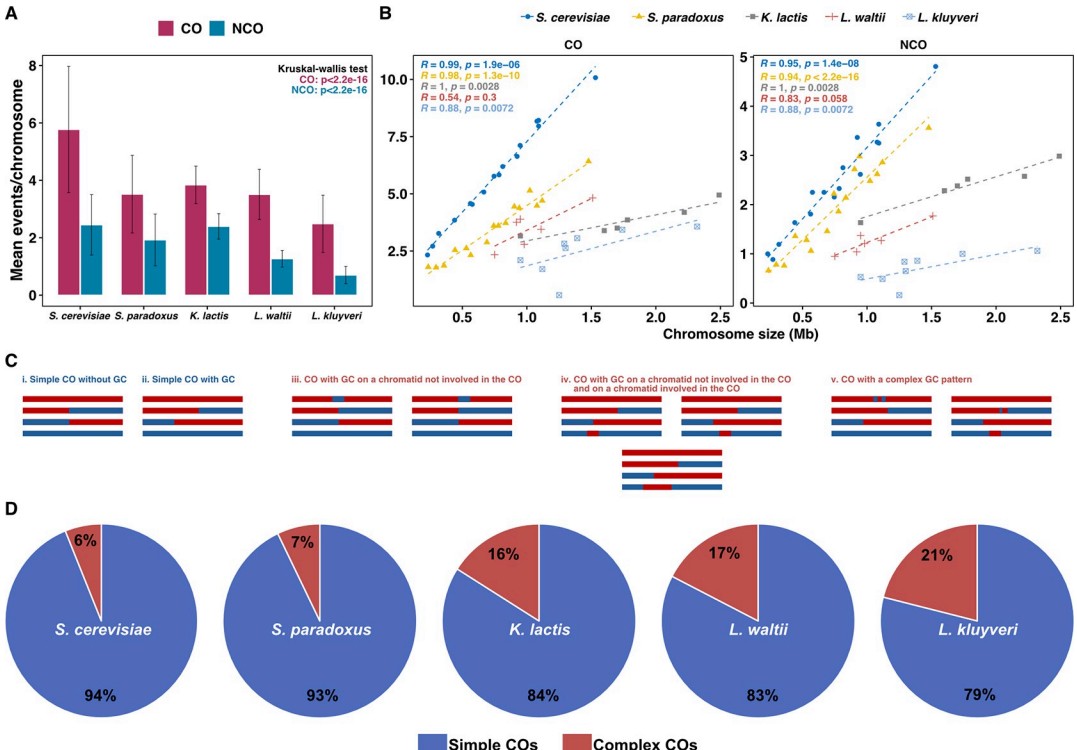

**Fig 2. Summary description of detected events. A.** Barplot depicting frequencies of crossovers (CO) and noncrossovers (NCO) per chromosome per meiosis for *S. cerevisiae* (n = 72), *S. paradoxus* (n = 59), *K. lactis* (n = 205), *L. waltii* (n = 192) and *L. kluyveri* (n = 59) [n-total number of tetrads analyzed for each species]. Bars represent standard deviation. **B.** Scatter plot of mean crossover and noncrossover counts per chromosome versus chromosome size for *S. cerevisiae*, *S. paradoxus*, *K. lactis*, *L. waltii* and *L. kluyveri* (R, Pearson's correlation). Best fit line obtained through linear regression. **C.** Cartoons representing CO events as defined in Anderson et al. 2011 [81]. **i** and **ii.** Simple CO with or without GC, with their midpoints not within 5 kb of another CO or NCO; **iii.** CO with a GC tract on a chromatid not involved in the CO; **iv.** CO with two GCs on a chromatid involved in the CO and on a chromatid not involved in the CO; **v.** CO with complex (discontinuous) GC. **D.** Pie chart representing the fraction of simple (Type 0 and 1) and complex (Type 2, 3, 4 and 8) crossovers across the five species. The complex COs were increased in non-*Saccharomyces* yeasts (*K. lactis*, *L. waltii* and *L. kluyveri*) compared to *Saccharomyces* yeasts (*S. cerevisiae* and *S. paradoxus*) (Wilcoxon test, p<0.01). The events have been classified as described in Anderson et al. 2011 [81].

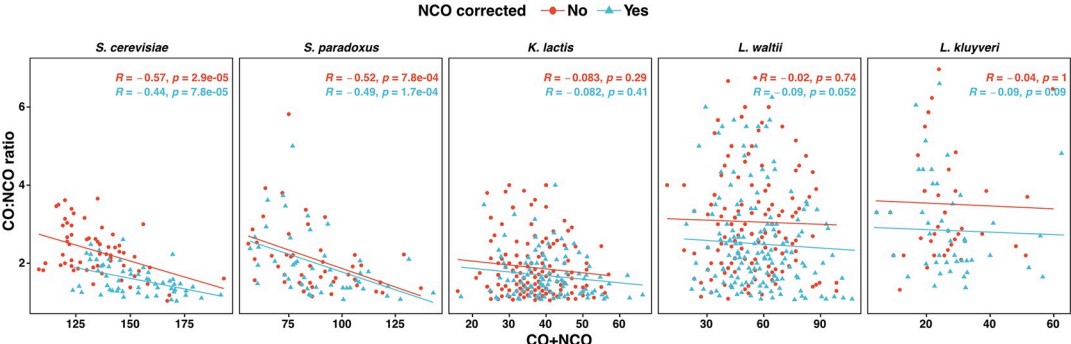

**Fig 3. Loss of crossover homeostasis in non-*Saccharomyces* yeasts (*K. lactis*, *L. waltii* and *L. kluyveri*) compared to *Saccharomyces* yeasts (*S. cerevisiae* and *S. paradoxus*).** Scatter plot depicting the relationship between the CO:NCO ratio and total interhomolog (CO+NCO) events per meiosis (R, Pearson's correlation). Best fit line obtained through linear regression. Average CO:NCO ratios (not corrected/corrected) were 2.3/1.8, 2.0/1.8, 1.8/1.5, 2.5/2.2 and 2.6/2.5 respectively for *S. cerevisiae*, *S. paradoxus*, *K. lactis*, *L. waltii* and *L. kluyveri*. The corrected NCO frequency was estimated as described in Mancera et al. 2008 [50].

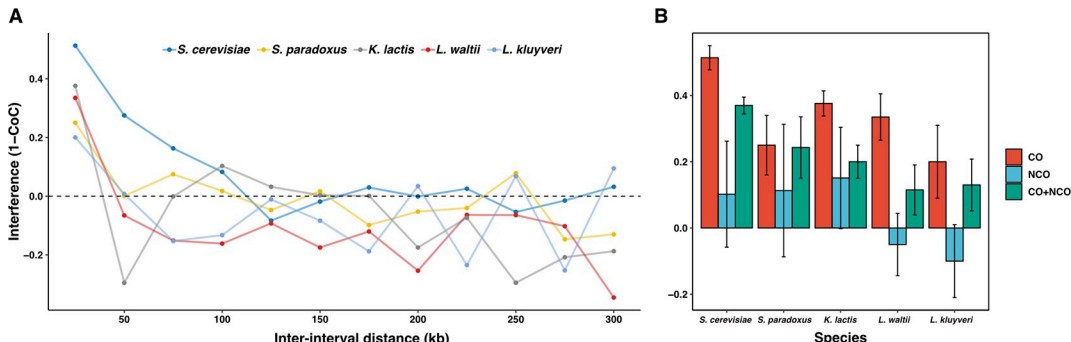

**Fig 4. Altered crossover interference across the yeast species. A.** Interference (1 –CoC) for crossovers in *S. cerevisiae*, *S. paradoxus*, *L. waltii*, *K. lactis* and *L. kluyveri* respectively. The CoC was determined for each inter-interval distance for every possible interval pair across the genome and the average is plotted. **B.** Interference calculated as 1-CoC for a bin size and inter-interval distance of 25 kb is shown for COs only, NCOs only, or all events (CO+NCO) across the species.

(K-S test, p < 0.0005). Altogether, these observations reveal the existence of CO interference in *K. lactis*, but its strength is weak and resembles that observed in *L. waltii*, which lacks most ZMM components. As for *L. waltii*, these properties suggest that the source of CO interference in *K. lactis* results from the patterning of CO precursors.

## COs hotspots and coldspots in *K. lactis*

To identify CO hotspots and coldspots, we used the 4,702 single COs (439 COs per Mb) to determine the local recombination rate (Fig 1, S3 Table). CO hotspots were identified using a 5 kb window size, and their significance was evaluated by permutation analysis ($10^5$ permutations, FDR of 2%), following a similar methodology used in previous studies in *L. kluyveri* and *L. waltii* [39,43]. CO coldspots were identified using 20 kb windows to improve detection sensitivity. In total, we detected 39 CO hotspots containing an average of 12 COs, and 31 CO coldspots deprived of any CO (Fig 1, S4 Table).

None of the CO hotspots detected in *K. lactis* were shared with *S. cerevisiae*, *L. waltii* and *L. kluyveri*, and vice versa. None of the CO hotspots were enriched for any functional class of genes. Similar to other yeast species, CO were overall depleted around centromeres but only *CEN6* overlapped is included in a CO coldspot (Fig 1). In addition, CO coldspots of *K. lactis* are enriched at chromosome ends: 9/12 chromosome ends are CO cold spots, and 2/3 chromosome ends that are not CO cold spots show very weak CO frequency nearby, including two CO cold spots. Overall, this suggests an enrichment of CO around centromeric regions and a depletion near chromosome ends. The *MAT* locus is a CO cold spot in *K. lactis*, as it is in many yeasts and fungi [54]. In *K. lactis*, recombination suppression around the *MAT* locus extends upto 50 kb, which is similar to the patterns observed in *S. cerevisiae*, *S. paradoxus*, and *L. waltii*. However, in *L. kluyveri*, recombination suppression around the *MAT* locus extends over the entire chromosome arm [43,44]. We found that CO coldspots were significantly associated with DNA repair genes (p<0.05; Hypergeometric test). Such genes include orthologs of the structure specific nuclease encoding gene *MUS81* (KLLA0E03015g).

Although centromeres are depleted in CO, every chromosome shows at least one CO hotspot within 100 kb next to its centromere, a property reminiscent of *A. thaliana* CO distribution, although at a different scale [55,56]. Overall, the density of CO is higher around centromeres than around chromosome ends in *K. lactis*. As a consequence, the density of CO hotspots is the highest on the smallest chromosome, which interestingly also exhibits the highest number of CO hotspots.

## Variable repertoire of recombination events in budding yeasts

To better understand the regulation of meiotic recombination and its evolution across species, we conducted a comparative analysis of the recombination landscapes among the five budding yeast species for which such data are available, namely *S. cerevisiae*, *S. paradoxus*, *L. kluyveri*, *L. waltii*, and *K. lactis* [12,39,43,57]. In addition to the dataset of Brion et al. 2017 [43], we sequenced and analyzed an additional 10 tetrads of *L. kluyveri*. The *Lachancea* and *Kluyveromyces* species branched from the *Saccharomyces* lineage more than 100 million years ago, before to the ancestral whole-genome duplication (WGD) event specific of the *S. cerevisiae* lineage (S8 Fig) [58]. The WGD event changed the basal chromosome numbers from 8 to 16. However, in post-WGD species, the chromosome number varies between 10 and 16, whereas non-WGD species have chromosome numbers ranging from 6 to 8 [59].

We cumulatively analyzed 577 meioses, comprising 19, 212 COs and 9, 939 NCOs for the five species [12,39,43,57]. The intra-specific hybrids used to generate such data show slight differences in genetic divergence and spore viability, *i.e. S. cerevisiae* (0.5%, 85%), *S. paradoxus* (1.1%, 64%), *L. kluyveri* (0.5%, 70%), *L. waltii* (0.6%, 71%), and *K. lactis* species (0.3%, 96%) (Table 1). Similar to previous findings in *S. cerevisiae* hybrids, the meiotic spore viability here also exhibits a strong negative correlation with hybrid divergence ($R^2$ = -0.72, p<0.01) [60,61]. Note that while CO detection is optimal using such genome-wide sequencing strategies, NCO counts maybe underestimated because of detection issues including low local SNP density and detection thresholds which may differ within these five datasets.

Among these five species, the CO and NCO frequencies per meiosis show at least a 5-fold and 7.5-fold difference, respectively, with the *Saccharomyces* species having more CO and NCO than non-*Saccharomyces* species, which correlates with about twice more chromosomes (Fig 2A, Table 1). *S. cerevisiae* has more CO and NCO than *S. paradoxus*, which could result from either more Spo11-DSBs per meiosis and/or an increased inter-homolog bias in *S. cerevisiae* compared to *S. paradoxus*. Conversely, *L. kluyveri* has the lowest CO and NCO number per chromosome per meiosis, which could result from a lower number of Spo11-DSBs per meiosis and/or a lower inter-homolog bias than any of the other four species.

Despite this low recombination rate, *L. kluyveri* has the highest CO:NCO ratio (2.6) suggesting a bias toward CO formation at inter-homolog recombination sites in this species. Similarly, *L. waltii* has more CO but less NCO per chromosome per meiosis (CO:NCO ratio 2.5) than *K. lactis*, also suggesting a bias toward CO formation more pronounced in *L. waltii* than in *K. lactis*.

The average frequency of CO and NCO per chromosome shows a linear relationship with chromosome size across species (Figs 2B–2C and S6A). However, the linear fit is very good for the Saccharomyces species, but it is worse for the non-Saccharomyces species. CO density shows a negative correlation with chromosome size in S. cerevisiae and S. paradoxus but not in K. lactis or the Lachancea species (S6B Fig). This result underscores a non-random distribution of recombination events, specifically enriching the small chromosomes in Saccharomyces yeasts only.

The average conversion tract lengths associated with CO (GCco) ranged from 1.8 kb to 2.5 kb, and that of NCO-associated tracts ranged from 1.2 kb to 2.2 kb (S3 Table, S5 Fig). These conversion events resulted in the conversion of 0.9% to 1.3% of the markers per meiosis across all species, except for *S. cerevisiae* where the proportion was slightly higher (2.0%) (p<0.01; Wilcoxon test).

In total 62–87% of the COs were associated with a detectable gene conversion tract in the different species (S3 Table). The fraction of CO with and without detectable conversion tracts are similar for the five species (p>0.05; Wilcoxon test). Interestingly, we observed at least a

2.5-fold increase in complex CO events involving more than two chromatids in *L. kluyveri*, *L. waltii* and *K. lactis* (Fig 2C–2D), pointing at more frequent complex recombination intermediates in these species.

Previous studies in *S. cerevisiae* and *S. paradoxus* have shown that CO and NCO events are positioned further away from the centromeres than chromosome ends. For the non-*Saccharomyces* species, we observed that COs are positioned further away from chromosome ends than centromeres (S9 Fig) [50,62]. However, it is important to note that the overall distances of CO events from centromeres and chromosome ends in *L. waltii*, *L. kluyveri* and *K. lactis* are significantly greater than in *S. cerevisiae* or *S. paradoxus* (S9 Fig).

## *L. waltii*, *L. kluyveri* and *K. lactis* show only residual CO interference compared to *Saccharomyces* species

We assessed crossover interference by modeling inter-crossover distances as a gamma distribution (Table 2; S7 Fig). To facilitate comparisons between *Saccharomyces* and *non-Saccharomyces* yeasts exhibiting variable crossover frequencies, we converted the inter-crossover distances from physical units into genetic distances (cM) [12,63]. The median inter CO distance in *S.cerevisiae* was 37.1 cM, similar to previous estimates [12]. The median inter-CO distances for *S. paradoxus*, *L. waltii*, *K. lactis* and *L. kluyveri* were 31.2 cM, 21.9 cM, 22 cM, and 14.9 cM, respectively. The reduced inter-CO distances in these species relative to *S. cerevisiae* could indicate reduced CO interference, notably for the non-*Saccharomyces* species (Table 2, S10A Fig). Fitting inter-CO distances with a gamma distribution reveals γ-values statistically different from a random distribution for the five species studied (p<0.05; Kolmogorov-Smirnov test) (Table 2, S7 Fig). The estimated γ-value in *S. cerevisiae* was 1.89 similar to previous estimates [12,50,53,64]. *S. paradoxus* showed an intermediate γ-value of 1.54 and the three non-*Saccharomyces* species showed the lowest and comparable γ-values (Table 2). We did not detect chromatid interference in either of these species [64].

Next, using the CoC method, we detected that CO interference spreads over more than 100 kb and 50 kb in *S. cerevisiae* and *S. paradoxus*, respectively, compared to 50 kb or less in *L. kluyveri*, *L. waltii* and *K. lactis* (p<0.01; Chi square test) (Fig 4A). We also observed that non-crossover (NCO) events do not exhibit interference, while the total recombination events (CO+NCO) still show residual CO interference. (Fig 4B). It was proposed that CO homeostasis is a consequence of CO interference [11,22]. The negative correlation between the CO:NCO ratio

**Table 2. Summary of the CO control metrics across the five species.**

| Species | Missing ZMM genes | γ-value | Median Inter-CO distance | CoC >0 | CO homeostasis | CO variance vs Mean CO[#] | %E0 | Observed vs expected $E_0$[##] |
|---|---|---|---|---|---|---|---|---|
| *S. cerevisiae* | - | 1.89 | 37.1 cM | >100 kb | Yes | Under dispersed (p<0.05) | 3% | Biased (p<0.01) |
| *S. paradoxus* | - | 1.54 | 31.2 cM | >50 kb | Yes | Under dispersed (p<0.05) | 62% | Random (p>0.01) |
| *K. lactis* | SPO16; MSH5 | 1.29 | 22 cM | <50 kb | No | Random (p>0.05) | 12% | Random (p>0.01) |
| *L. waltii* | ZIP2-4; SPO16; MSH4-5 | 1.24 | 21.9 cM | <50 kb | No | Random (p>0.05) | 31% | Random (p>0.01) |
| *L. kluyveri* | - | 1.21 | 14.9 cM | <50 kb | No | Random (p>0.05) | 55% | Random (p>0.01) |

# Chi-square test p-values for observed variance vs expected variance from a random distribution of COs

## Chi-square test p-values for observed $E_0$s vs expected $E_0$s from a random distribution of COs

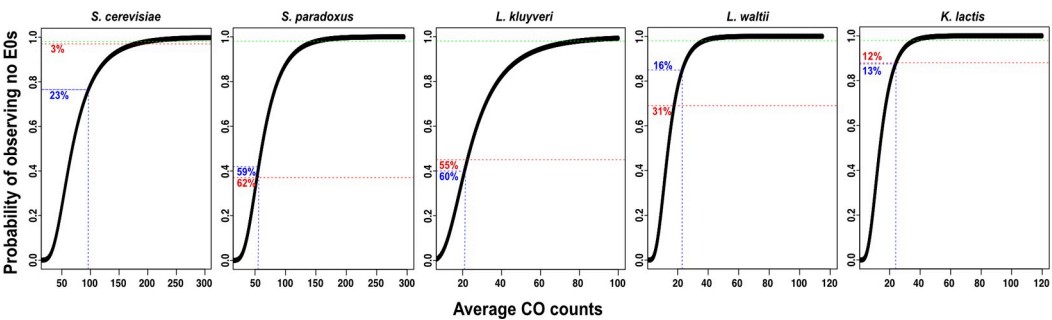

**Fig 5. Probability of observing no non-exchange chromosomes (E$_0$s) in the five species.** The probability of observing no E0 is plotted against the average CO number distributed randomly among the chromosomes (Poisson distribution). Numbers in red and blue indicate the observed and expected percentages of meiosis with non-exchange chromosomes, respectively.

and the total number of recombination events (CO+NCO) per tetrad that characterizes CO homeostasis is observed in *S. cerevisiae* (r = -0.57; p<0.01) and *S. paradoxus* (r = -0.52; p<0.01), but not in the non-*Saccharomyces* species (Fig 3).

To further evaluate the randomness of the CO distributions, we plotted the variance of CO counts against the corresponding mean CO values for each chromosome since under a Poisson distribution that assumes that CO events are independent of each other, the variance should equal the mean. We observed an under-dispersion of the CO counts across all chromosomes within *S. cerevisiae* and, to a lesser extent *S. paradoxus* (p<0.05; Chi square test) (S10B Fig) [65]. The reduced variability in CO counts suggests the presence of mechanism(s) controlling CO distribution in a non-random manner. Conversely, we did not observe such an under-dispersion of CO counts across all chromosomes in the non-*Saccharomyces* species (p>0.05; Chi square test) (Table 2, S10B Fig).

Altogether, these results suggest that CO interference is present only in the *Saccharomyces* species, and that the CO interference signal in the non-*Saccharomyces* species only comes from the CO precursors patterning as in *L. waltii* [39]. This result is particularly striking for *L. kluyveri* that contains all the ZMM genes known to generate interfering CO [43].

## Frequent non-exchange (E$_0$) chromosomes except in *Saccharomyces cerevisiae*

The frequency of meioses with 1 or >1 E$_0$/non-exchange chromosomes is 3% in *S. cerevisiae*, 62% in *S. paradoxus*, 55% in *L. kluyveri*, 31% in *L. waltii* and 12% in *K. lactis*, and all species except *S. cerevisiae* display tetrads with >1 non-exchange chromosome (Table 2). Surprisingly, *S. paradoxus* displayed the highest frequency of E$_0$ chromosomes despite having a higher CO rate than *L. kluyveri*, *L. waltii* and *K. lactis*, and showing positive CO interference.

We next calculated the expected percentage of meioses with non-exchange chromosome for the five species assuming the CO count per chromosome follows per meiosis a Poisson distribution (Fig 5). The observed and expected frequency of E$_0$ chromosomes in *S. paradoxus* (62% and 59%), *L. kluyveri* (55% and 60%), *L. waltii* (31% and 16%) and *K. lactis* (12% and 13%) were similar. This is compatible with close to random CO distributions in these species (p>0.05; Chi square test). However, for *S. cerevisiae*, the observed percentage of meioses involving non-exchange chromosomes is significantly lower (3%) than expected (23%), supporting a non-random distribution of CO (p<0.01; Chi square test).

Despite this non-random CO distribution and lower achiasmatic chromosome frequency than expected, small chromosomes are the most frequently achiasmatic in *S. cerevisiae* [12,63].

Medium and large chromosomes become achiasmatic only in mutants showing CO formation defects [63]. In *S. paradoxus*, which displays a 1.6-fold lower CO rate compared to *S. cerevisiae* (Table 1; Fig 2A), small and medium sized chromosomes are the most frequently $E_0$s (95%), followed by large chromosomes (5%). For the three non-*Saccharomyces* species, the association between chromosome size and $E_0$s is less apparent. However, their chromosomes are on average much larger than *Saccharomyces* chromosomes, with their smallest chromosomes being close to 1 Mb long, which is equivalent to medium-large chromosomes of *Saccharomyces* species (S2 Table).

Finally, we monitored the presence of gene conversions on $E_0$ chromosomes to determine if these chromosomes underwent inter-homolog interactions. In *S. cerevisiae* and *S. paradoxus*, all $E_0$ chromosomes exhibit at least one gene conversion event (S2 Table), suggesting that they failed to convert any inter-homolog interactions into a CO. In contrast, 85% (23), 49% (42) and 20% (5) of the non-exchange chromosomes were devoid of NCO in *L. kluyveri*, *L. waltii*, and *K. lactis*, respectively (S2 Table). This shows that, in these non-*Saccharomyces* species, a large fraction of the non-exchange chromosomes did not undergo any inter-homolog interaction at all.

## ZMM genes display elevated evolutionary rates

Variants in genes controlling meiotic recombination such as *MER3*, *MLH3*, *MSH4-5*, *ZIP3*, have been identified as causative for recombination rate variation [66–68]. Moreover, meiosis proteins in some pathogenic yeast species exhibit increased divergence, and such increased evolutionary rates in protein sequences have been linked to their abandonment [69,70].

To estimate the evolutionary rates of the ZMM genes, we calculated the mean pairwise dN/dS ratios for all genes present in species with available population genomic datasets: *S. cerevisiae* (1,011 isolates), *L. kluyveri* (28 isolates), and *K. lactis* (41 isolates). We focused on genes having orthologs in all of the three species (n = 3,156) [48,71,72]. Additionally, we identified meiosis genes as those annotated with the gene ontology annotation 'meiosis' in *S. cerevisiae* (n = 190) (S7 Table). A higher dN/dS ratio than expectation is typically interpreted as a stronger amount of directional selection *i.e.*, more functional substitutions per non-functional substitution than expected. The dN/dS ratios for meiosis genes, in comparison to the rest of the genes, were elevated in all three species, with significance detected in *S. cerevisiae* and *L. kluyveri* (Fig 6, S7 Table). Interestingly, we detected an enrichment of meiosis genes, specifically those implicated in crossing over during meiosis I, among genes exhibiting dN/dS ratios greater than the genome-wide median across all three species (p = $3.06 \times 10^{-5}$, Hypergeometric test).

Among the meiosis genes, the ZMM genes consistently exhibited higher dN/dS ratios compared to the rest of the meiosis genes in all three species (p<0.05; Wilcoxon test). In fact, *ZIP2* and *SPO16* are amongst the most diverged genes in the *S. cerevisiae* and *L. kluyveri* genomes with dN/dS ratios in the 96% percentile. The elevated dN/dS ratio observed in ZMM genes may be attributed to higher mutation rates, as evidenced by higher $\pi$ (nucleotide diversity) values for ZMM genes compared to the rest of the genome. Notably, this increase in $\pi$ values was significant in *L. kluyveri* and *K. lactis* but not in *S. cerevisiae* (S7 Table). In summary, these findings suggest that ZMM genes experience less stringent evolutionary constraints, but for which the selective advantage is still under debate [73].

## Discussion

Studying fine-scale variations in recombination rates between closely related species can offer insights into the evolutionary forces influencing recombination, providing a window on the

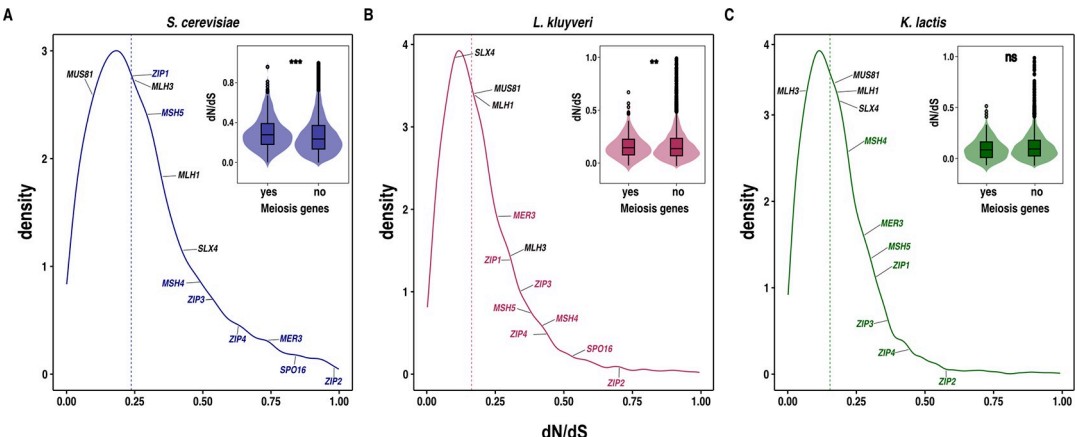

**Fig 6. Elevated evolutionary rates in ZMM genes. A-C.** Distribution of dN/dS ratios for all the genes across *S. cerevisiae*, *L. kluyveri* and *K. lactis* species respectively. ZMM genes are indicated in the distribution. **Inset:** Violin plot comparing the dN/dS ratios of meiosis genes with the rest of genes that have orthologs in all the three species. Hollow circles on the violin plot represent ZMM genes.

dynamic processes that shape genomes over time [42,74,75]. In this context, we characterized the recombination landscape of an additional yeast species, *K. lactis*, and subsequently performed a comparative analysis of the recombination landscapes of five species of the Saccharomycotina subphylum: *S. cerevisiae*, *S. paradoxus*, *L. kluyveri*, *L. waltii* and *K. lactis* [12,39,43,50,57,62,63]. Overall, we found substantial variations in the meiotic recombination landscapes and properties of these five species, some of which may call for experimental testing. Both the CO and NCO rates are higher in the *Saccharomyces* yeasts than in the non-*Saccharomyces* species. This could result from a higher number of Spo11-DSB and/or a stronger bias to recombine with the homolog in the *Saccharomyces* species. Another main difference is the larger extent of crossover interference in *Saccharomyces* species compared to non-*Saccharomyces* species. While it is expected that *L. waltii* and *K. lactis* show residual CO interference due an incomplete ZMM complement, the weak CO interference in *L. kluyveri* is surprising since all the ZMM players are present. Overall, this work adds two budding yeast species, *K. lactis* and *L. kluyveri*, to the list of species that lost the bona fide meiotic crossover interference, underlying the dispensability of this pathway, at least within budding yeasts.

Interestingly, our comparative analysis of meiotic recombination revealed that only *S. cerevisiae* exhibits the three main components of CO control, *i.e.*, ZMM-mediated CO interference, CO homeostasis and CO assurance. Although weaker than in *S. cerevisiae*, *S. paradoxus* shows CO interference (γ-value of 1.89 vs 1.54, respectively), it also shows CO homeostasis, but it shows the highest frequency of achiasmate chromosomes (62% of meioses). This suggests that the mechanism responsible for the obligatory CO is poorly efficient in *S. paradoxus*, contrarily to its close relative *S. cerevisiae* where achiasmate chromosomes are rare. Alternatively, but not exclusively, the high frequency of achiasmate chromosomes in *S. paradoxus* could be a consequence of the higher frequency of SNPs in the hybrid used to generate the recombination map, since sequence polymorphisms can trigger the anti-recombinogenic activity of the mismatch repair machinery [51,76,77]. Overall, the fact that *S. paradoxus* retained CO interference and CO homeostasis but not CO assurance is a nice biological support of the uncoupling of CO interference and CO homeostasis from CO assurance [22]. It is also important to note that non-exchange chromosomes are also rather frequent in the non-*Saccharomyces* species, suggesting that these E0 chromosomes can be segregated rather efficiently in *Saccharomyces*, *Kluyveromyces* and *Lachancea* species. Altogether, our results

highlight that the non-*Saccharomyces* species lack CO homeostasis, and as expected from a random distribution of CO, they show a variance of the CO number similar to the mean CO number per meiosis, and they show frequent achiasmate chromosomes. In addition, these species show close to random CO distributions, hence lack CO interference as well. For *L. waltii* and *K. lactis*, this agrees with a dysfunctional ZMM pathway. Conversely, *L. kluyveri* despite harboring a complete complement of ZMM pathway genes also lacks CO homeostasis, suggesting the improper functioning of the ZMM pathway.

The structure of the recombination events can inform on the recombination mechanism itself. In the non-*Saccharomyces* species, COs show more frequently a conversion tract on a third chromatid not involved in the CO itself compared to the *Saccharomyces* species. These could arise from template switching during recombination, involving both non-sister and sister chromatids in a single repair event [8,51,78]. In *S. cerevisiae*, the frequency of such events increases in the absence of the Sgs1 helicase [8,26,33], and in the combined absence of Mlh2 and Msh5 [8]. The absence of *MSH4/5* could explain at least part this phenotype in *L. waltii*. It could also apply to *K. lactis* since its *MSH5* gene is frameshifted. Alternatively, but not exclusively, the complex conversions observed in the non-*Saccharomyces* species could also result from different properties of the Sgs1 helicase on recombination intermediates compared to *S. cerevisiae*.

In *S. cerevisiae*, NCO events that look like "apparent double CO" have been used to infer a lack of CO bias during the resolution of recombination intermediates, such as in the *zip3* and *sgs1* mutants [57]. We did not find evidence of increase in such scars in the NCO tracts in *L. waltii* and *K. lactis* that likely have compromised ZMM pathways. This could suggest that the resolution of recombination intermediates is also biased toward CO in these species independently of the ZMM pathway. However, recombination intermediates resolution toward NCO does not exclusively yield apparent double CO but also yields simple NCO tracts. This prevents us to conclude about the CO biased resolution of recombination intermediates in the ZMM deficient *L. waltii* and *K. lactis* species and calls for experimental testing.

Recombination hotspots between *S. cerevisiae* and *S. paradoxus* are well conserved, likely due to their highly collinear genomes and similar control of crossover (CO) patterning [62,79,80]. In contrast, recombination hotspots in *L. kluyveri*, *L. waltii*, and *K. lactis* display little to no conservation [39,43]. This lack of conservation may be attributed to structural variations that affect crossover strength in cis as discussed in [43].

## Materials and methods

### Yeast strains and growth conditions

The *K. lactis* isolates used in this study are described in S4 Table. Strains were grown in standard YPD medium or YPD supplemented with hygromycin (300 mg/ml) or nourseothricin (100 mg/ml) and agar 20 g/l for solid medium at 30˚C. Sporulation was performed at 30˚C on 1% potassium acetate agar plates. The *L. kluyveri* hybrid used in this study is as in [43]. Note that sporulation of *L. kluyveri* was in liquid 1% potassium acetate and that tetrads were dissected after 24 hours only on sporulation medium to prevent as much as possible return to growth or intra-tetrad mating, hence the presence of loss of heterozygosity blocks in the final hybrid.

### Mating, sporulation, and spore isolation

Stable haploid isolates YJS5019 (UCD67-376 or C6) and YJS5035 (UCD72-212 or H7) were mixed on YPD plates overnight at 30˚C. Double resistant diploid single colonies were isolated from YPD supplemented with hygromycin and nourseothericin after 48 hours. Diploidy of the

hybrid was confirmed via flow cytometry. After ploidy validation, one hybrid diploid was selected and sporulated for 2–3 days on 1% potassium acetate plates at 30˚C. Tetrads dissections were performed using the SporePlay (Singer Instrument) without any pre-treatment. Dissection of about 1,000 tetrads showed 2:2 segregation of the *HphMX* and *NatMX* markers.

## Illumina high-throughput sequencing

Total genomic DNA was extracted using the 96-well E-Z 96 Tissue DNA kit (Omega) following a modified bacterial protocol [39]. DNA concentration was measured using the Qubit dsDNA HS assay (ThermoFischer) and the fluorescent plate-reader TECAN Infinite Pro200 and DNA quality was evaluated using a NanoDrop 1000 Spectrophotometer (ThermoFischer).

DNA libraries were prepared from 5 ng of total genomic DNA using the NEBNext Ultra II FS DNA Library kit for Illumina (New England Biolabs). All volumes specified in the manufacturer's protocol were divided by four. The adaptor-ligated DNA fragments of about 300-bp were amplified with 8 cycles of PCR using indexed primers. A combination of 48 i7 oligos (NEBNext Multiplex Oligos for Illumina, NEB) and 24 i5 oligos (Microsynth) were designed enabling multiplexing up to 1152-samples. After quality check using a Bioanalyzer 2100 (Agilent Technologies) and quantification using the Qubit dsDNA HS assay, 4 nM of each of the 782 libraries were pooled and run on a NextSeq 500 sequencer with paired-end 75 bp reads by the EMBL's Genomics Core Facility (Heidelberg, Germany).

## Oxford Nanopore sequencing

DNA extraction was performed using the Monarch HMW DNA extraction kit (NEB) following the manufacturers protocol. DNA concentration was measured using the Qubit dsDNA HS assay (ThermoFischer) and DNA quality was evaluated using a Nanodrop 1000 Spectrophotometer (ThermoFischer). Library preparation was done using Oxford Nanopore's ligation sequencing kit (SQK-LSK 109) coupled with EXP-NBD 104 native barcoding kit. Sequencing was performed on a MinIon R9.4.1 flowcell.

## Genome assembly and SV detection

Base calling was carried out using Guppy v.6.3.8 with the SUP model. Porechop (v.0.2.4) was used to process the raw fastq files in order to eliminate any residual adapters and barcodes. The best reads (—min_length 1000 -target_bases 550000000—length_weight 10) were down sampled and saved for our assembly using filtlong (v.0.2.1). Next, we used Flye (v.2.8.1) and NECAT (v. 0.0.1_update20200803) to assemble the genome. Subsequently, two rounds of polishing were performed on these assemblies using Medaka (model r941_min_sup_g507) and the associated Nanopore readings. Racon (1.5.0) was used to further refine genomes using previously obtained Illumina short reads. After polishing these genomes, we ran MUMandCo (v.3.8) with the -g 10700000 option to find structural variations in our assemblies.

## Mapping and single Nucleotide Polymorphisms (SNPs) calling

Illumina sequencing reads were mapped to the YJS5035 genome assembly previously described using bwa mem (v0.7.17). The resulting bam files were sorted and indexed using SAMtools (v1.9). Duplicated reads were marked, and sample names were assigned using Picard (v2.18.14). GATK (v3.7.0) was used to realign remaining reads. Variants were then called using GATK UnifiedGenotyper.

## Segregation analysis

Following variant calling, SNPs called in the YJS5019 (C6) and YJS5035 (H7) were first filtered (bcftools view, v1.9) to define a set of confident markers, corresponding to positions with a single alternate allele, supported by at least ten sequencing reads in each parent and with >90% of the sequencing reads covering either the reference or alternate allele. For each segregant resulting from the hybrid, SNPs located at marker positions were extracted and parental origin was assigned based on SNP correspondence between haploid parents and spores at those positions. In this case, since YJS5035 is a parent as well as the reference, genotype and parental origin can actually be considered similar.

## Recombination events calling

After the extraction of SNP information in tetrads as well as assignment of parental origin, the information was summarized in segregation files and used as input for the CrossOver python script (Recombine suite) [81], using default parameters and adapted chromosome size/coordinates to fit the newly assembled reference of strain YJS5035 (H7).

## CO hotpots and coldspots detection and conservation analysis

We calculated the number of CO in 5 kb windows for hotspots and 20 kb windows for coldspots. To determine if the CO in a window deviated from chance expectations, we employed a permutation test to establish significance thresholds. We simulated the CO density per window through $10^5$ random distributions of 4,702 CO, adhering to a binomial distribution with equal probabilities for each event to occur within an interval. Within these simulations, the highest or lowest CO count per interval was extracted, and the 2000th highest or lowest value was used as the threshold to categorize hotspots and coldspots, respectively. Hotspots were identified as having a significance threshold of >12 CO per window, while coldspots were defined with a significance threshold of 0 CO per window.

The synteny between CO hotspots of *S. cerevisiae* [50], *L. waltii* [39], *L. kluyveri* [43] and *K. lactis* (this study) were tested. For each hotspot, the five closest genes were identified, and their potential orthologs in the other species genome were detected using the annotated genome of the reference strains and the *K. lactis* H7 reference sequence we assembled. If more than two orthologs linked to the same hotspot were found within a region less than 10 kb apart, the synteny was considered conserved and suitable for hotspot comparison. Hotspots were considered to be conserved if they were separated by less than 5 kb. We retrieved conserved syntenic blocks in *L. kluyveri*, *L. waltii* and *K. lactis* genomes containing at least two *S. cerevisiae* orthologs associated with one hotspot. *L. waltii* shares only five out of the 92 *S. cerevisiae* crossover hotspots (*RHO5*, *SLS1*, *GYP6*, *OLE1* and *MRPL8*), while *L. kluyveri* shares only one. *L. waltii* and *L. kluyveri* share no crossover hotspots (S11 Fig). In addition, we did not detect any of *K. lactis* hotspot associated with any of the other species.

## Detection of CO interference

We computed inter-crossover distance as the physical distance between successive crossovers. These measurements were subsequently transformed into genetic distances (cM). These values were converted into genetic distance using the formula: 1 Morgan = Genome size (in bp) × 2/ mean crossovers. The distribution of all inter-CO distances in the five species were fitted with a gamma law using the fitdistr function in R's MASS package, from which the shape (γ) and scale (θ) parameters of the fitted function were obtained. CO interference was assessed through the execution of a Kolmogorov-Smirnov test between the distribution of all inter-CO

distances and a gamma law with a shape of 1 (*i.e.*, lacking interference). The γ-value estimates for all species were consistent with previous estimates, except for *L. kluyveri* [43]. This discrepancy is attributed to our exclusive use of single meiotic CO here, yielding a γ-value of 1.21, while we previously included RTG-induced CO yielding a γ-value of 1.47 [43]. In both cases this γ-value is lower than that of *S. cerevisiae*. In addition, γ-value in *L. kluyveri* tetrads devoid of any LOH events (n = 11) was 1.2. The revised γ-value of *L. kluyveri* across all tetrads is now similar to that of *L. waltii* that is devoid of the ZMM genes and for which the source of CO interference is likely the patterning of CO precursors [39].

For the Coefficient of Coincidence (CoC) analysis, we partitioned the genome into 25 kb bins. For each bin, we computed the frequency of crossover events and determined both the expected and observed frequencies of double crossovers across all bin pairs. The CoC, represented as the ratio of observed crossover frequency to expected crossover frequency, was analyzed for all bin pairs with non-zero expected frequency, separated by a specified interval (in kilobases), and the results were averaged. The statistical significance of observed and expected double crossover frequencies was assessed using the chi-square test.

## Supporting information

**S1 Table. A)** Seven scaffolds representing the six *K. lactis* chromosomes and the mitochondrial genome. **B)** Structural variants detected in the C6 and H7 parental genomes compared to the reference isolate.
(XLSX)

**S2 Table.** List of all Non exchange chromosomes across all tetrads. Tract type indicates the type of gene conversion event detected on the chromosome and the number of events [81].
(XLSX)

**S3 Table.** Crossover events detected by Recombine in *K. lactis*. *S. cerevisiae* data pooled from Mancera et al. 2008, Oke et al. 2014, Krishnaprasad et al. 2015 [12,50,57]; *L. kluyveri* data pooled from Brion et al. 2017 [43]; *L. waltii* data pooled from Dutreux et al. 2023 [39]; *S. paradoxus* data pooled from Liu et al. 2019 [62].
(XLSX)

**S4 Table. A)** List of crossover hotspots and coldspots in *K. lactis*. **B)** List of genes within the GO term "DNA repair". DNA repair genes associated with CO coldspots highlighted in blue.
(XLSX)

**S5 Table. Noncrossover events detected by Recombine in *K. lactis*. *S. cerevisiae* data pooled from Mancera et al.** 2008, Oke et al. 2014, Krishnaprasad et al. 2015 [12,50,57]; *L. kluyveri* data pooled from Brion et al. 2017 [43]; *L. waltii* data pooled from Dutreux et al. 2023 [39]; *S. paradoxus* data pooled from Liu et al. 2019 [62].
(XLSX)

**S6 Table. Sequence statistics of all the strains used in this study.**
(XLSX)

**S7 Table. dN/dS ratios and Pi values for *S. cerevisiae*, *L. kluyveri* and *K. lactis*.**
(XLSX)

**S1 Fig. A)** Population structure of *K. lactis* strains [48]. Haploids isolates used to generate the mapping population are highlighted in blue and red. Branch lengths are proportional to the number of sites that discriminate each pair of strains. **B)** Scheme of the experimental design.
(TIFF)

**S2 Fig. A)** The plot indicates the number of SV of each type w.r.t. the reference isolate (innermost ring). **B)** Reorganization of the C6 and H7 chromosomes as a function of the reference isolate chromosomes.
(TIFF)

**S3 Fig. A-C)** Dot plot representing the alignment of reference isolate chromosomes (x-axis) w.r.t. C6 and H7 respectively as well as between C6 and H7 respectively.
(TIFF)

**S4 Fig.** Distribution of inter marker distances within the diploid hybrid (CxH7). The median inter-marker distance is 187 bp.
(TIFF)

**S5 Fig.** Violin plot representing the CO and NCO associated tract sizes across the five species.
(TIFF)

**S6 Fig.** Spearman correlations between **A)** Total events (CO + NCO) per chromosome and **B)** Crossover density per chromosome with respect to chromosome size across the five species.
(TIFF)

**S7 Fig.** Inter-CO distances (cM) modelled as a gamma distribution across the five species.
(TIFF)

**S8 Fig.** Phylogenetic tree representing the divergence of the five species based on 26S rDNA sequences of published strains.
(TIFF)

**S9 Fig.** Violin plots representing average distances (in base pairs) of crossover from centromeres and chromosome ends across the four species.
(TIFF)

**S10 Fig. A)** Distribution of inter-CO distances (cM) across the five species. **B)** Variance of CO counts per chromosome plotted against the average CO counts per chromosome.
(TIFF)

**S11 Fig.** Density of crossovers along the genome using a 5 kb window in the *S. cerevisiae* genome (Mancera et al. 2008; Oke et al. 2014; Krishnaprasad et al. 2015 combined dataset) [12,50,57]. Horizontal dotted green line represents crossover hotspot significance threshold. Solid spheres represent the conserved CO hotspots from either *L. kluyveri* (red) *or L. waltii* (blue).
(TIFF)

## Acknowledgments

We thank the CRCM integrative bioinformatics (CIBI) platform for the analysis of the *L. kluyveri* tetrads.

## Author Contributions

**Conceptualization:** Bertrand Llorente, Joseph Schacherer.

**Data curation:** Abhishek Dutta, Fabien Dutreux, Anne Friedrich.

**Formal analysis:** Abhishek Dutta, Fabien Dutreux, Anne Friedrich, Gauthier Brach, Maxime Gaudin.

**Funding acquisition:** Bertrand Llorente, Joseph Schacherer.

**Investigation:** Fabien Dutreux, Marion Garin, Claudia Caradec, Gauthier Brach, Pia Thiele.

**Methodology:** Abhishek Dutta, Fabien Dutreux, Marion Garin, Anne Friedrich, Gauthier Brach, Maxime Gaudin.

**Project administration:** Bertrand Llorente, Joseph Schacherer.

**Resources:** Bertrand Llorente, Joseph Schacherer.

**Software:** Abhishek Dutta, Fabien Dutreux, Anne Friedrich, Gauthier Brach, Maxime Gaudin.

**Supervision:** Abhishek Dutta, Bertrand Llorente, Joseph Schacherer.

**Validation:** Abhishek Dutta.

**Visualization:** Abhishek Dutta.

**Writing – original draft:** Abhishek Dutta, Bertrand Llorente, Joseph Schacherer.

**Writing – review & editing:** Abhishek Dutta, Bertrand Llorente, Joseph Schacherer.

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
