## [Decision Letter · Decision Letter 0]

29 Aug 2024

Dear Dr Schacherer,

Thank you very much for submitting your Research Article entitled 'Multiple independent losses of crossover interference during yeast evolutionary history' to PLOS Genetics.

The manuscript was fully evaluated at the editorial level and by independent peer reviewers. The reviewers appreciated the attention to an important topic but identified some concerns that we ask you address in a revised manuscript.

We therefore ask you to modify the manuscript according to the review recommendations. Your revisions should address the specific point made by each reviewer 2 regarding inclusion of the numerical data used in the graphs.

To resubmit, log into your Editorial Manager account and select the option 'Revise Submission' in the 'Submissions Needing Revision' folder.

Yours sincerely,

Lorraine S. Symington

Academic Editor

PLOS Genetics

Giorgio Sirugo

Section Editor

PLOS Genetics

Reviewer's Responses to Questions

**Comments to the Authors:**

Reviewer #1: I reviewed this manuscript when submitted in Review Commons. The authors addressed all my minor comments properly.

As mentioned in my major comment, one issue I have in mind is the functions of ZMM genes in the meiosis of Kluyveromyces lactis (such as ZIP1, -2, -3, -4, MSH5 and MER3). Are these genes necessary for the formation of meiotic crossovers or not? If the authors’ idea is correct, it is unlikely that these ZMM genes promote the crossing-over. This could be simply tested by the construction of the deletion mutant of the genes (not complementation assay using S. cerevisiae) and checking the spore viability of the mutants as a minimum analysis of meiotic phenotypes. As in the response to the #1 reviewer, the authors showed some of these K. lactics genes are indeed induced during meiosis. This reinforces the idea of performing the analysis. But as I originally wrote, this is a bit out of the focus of the current version of this paper.

Reviewer #2: The authors have done an excellent job of addressing my previous review comments, and I support publication once the numerical data underlying graphs are included.

**Have all data underlying the figures and results presented in the manuscript been provided?**

Reviewer #1: Yes

Reviewer #2: **No: **Specific data for graphs in figures needs to be included.

PLOS authors have the option to publish the peer review history of their article (what does this mean?). If published, this will include your full peer review and any attached files.

Reviewer #1: **Yes: **Akira Shinohara

Reviewer #2: No

---

## [Editor Report · Decision Letter 1]

11 Sep 2024

Dear Dr Schacherer,

We are pleased to inform you that your manuscript entitled "Multiple independent losses of crossover interference during yeast evolutionary history" has been editorially accepted for publication in PLOS Genetics. Congratulations!

Yours sincerely,

Lorraine S. Symington

Academic Editor

PLOS Genetics

Giorgio Sirugo

Section Editor

PLOS Genetics

Comments from the reviewers (if applicable):

**Data Deposition**

http://datadryad.org/submit?journalID=pgenetics&manu=PGENETICS-D-24-00877R1

**Press Queries**

---

## [Editor Report · Acceptance letter]

21 Sep 2024

PGENETICS-D-24-00877R1 

Multiple independent losses of crossover interference during yeast evolutionary history 

Dear Dr Schacherer, 

We are pleased to inform you that your manuscript entitled "Multiple independent losses of crossover interference during yeast evolutionary history" has been formally accepted for publication in PLOS Genetics! Your manuscript is now with our production department and you will be notified of the publication date in due course.

With kind regards,

Zsofia Freund

PLOS Genetics

On behalf of:
